# Cardiovascular disease risk perception among community-dwelling adults in southwest Nigeria: A mixed-method study

**Oluwagbohunmi A. Awosoga**[1], **Olufemi O. Oyewole**[2,3]*, **Opeyemi M. Adegoke**[4], **Adesola C. Odole**[4], **Ogochukwu K. Onyeso**[1], **Chiedozie J. Alumona**[1,5], **Abiodun M. Adeoye**[6], **Happiness A. Aweto**[7], **Blessing S. Ige**[2], **Adetola C. Adebayo**[2], **Titilope L. Odunaiya**[8], **Grace M. Emmanuel**[2], **Nurudeen B. Sulaimon**[2], **Nse A. Odunaiya**[4]

**1** Faculty of Health Sciences, University of Lethbridge, Lethbridge, Alberta, Canada, **2** Department of Physiotherapy, Olabisi Onabanjo University Teaching Hospital, Sagamu, Nigeria, **3** College of Health Sciences, University of KwaZulu-Natal, Durban, South Africa, **4** Department of Physiotherapy, College of Medicine, University of Ibadan, Ibadan, Nigeria, **5** Department of Physiotherapy, College of Basic Medical Sciences, Chrisland University, Abeokuta, Nigeria, **6** Department of Medicine, College of Medicine, University of Ibadan, Ibadan, Nigeria, **7** Department of Physiotherapy, College of Medicine, University of Lagos, Lagos, Nigeria, **8** Department of Psychology, University of Ibadan, Ibadan, Nigeria

* oyewoleye@gmail.com, OyewoleO1@ukzn.ac.za

**Data Availability Statement:** The datasets analysed during the current study are available from the Open Science Framework open access

## Abstract

### Objective

The rising prevalence of cardiovascular diseases (CVD) remains a global concern. In Nigeria, the current prevalence of CVD was 76.11% with its attendance burden. The CVD risk perception of individuals is a precursor to the desired lifestyle modification necessary for CVD prevention and management. This study assessed the CVD risk perception and socio-demographic determinants among rural and urban dwellers in southwest Nigeria.

### Methods

The study employed a convergent parallel mixed-methods design involving concurrent data collection. The participants' CVD risk perception was obtained using the Perception of Risk of Heart Disease Scale (quantitative data) and a validated focus group discussion (FGD) guide (qualitative data). Quantitative analysis was completed using descriptive statistics, Phi, Cramer's V, and multivariate linear regression, while the FGD was thematically analysed.

### Results

The quantitative study involved 1,493 participants (62.4% women) with a mean age of 46.90 ±15.65 years, while the FGD involved 53 participants (52.8% women) with a mean age of 50.10±13.5 years. Over a quarter (28%) of the participants had a poor CVD risk perception; the mean score was 44.40±8.07. Rural residents had a significantly poorer CVD risk perception than their urban counterparts (Mean difference = -3.16, $p<0.001$). Having tertiary education ($\beta = 0.100$, $p < 0.001$), living in urban areas ($\beta = 0.174$, $p<0.001$), and living in Lagos ($\beta$

research repository (https://osf.io/7eq8w/?view_only=e54bf6241deb43af8a86e2e8396ed02d).

**Funding:** The study was funded by the Prentice Institute for Global Population and Economy, and the Faculty of Health Sciences, University of Lethbridge, Alberta, Canada. The funders had no role in study design, data collection and analysis, decision to publish, or preparation of the manuscript.

**Competing interests:** The authors declare that they have no competing interests.

= 0.074, $p$ = 0.013) and in Oyo, other than Ogun state (β = -0.156, $p$<0.001) significantly predicted having a good perception of CVD risk. FGD produced three themes: knowledge about CVD, CVD risk factors, and CVD prevention.

## Conclusion

Participants had a fair understanding of the causes and prevention of CVD. Yet, a substantial portion underestimated their own risk of developing CVD, particularly rural dwellers and people with lower education. More public health education is required to improve the CVD risk perception in southwestern Nigeria.

## Introduction

With advances in medical services and health campaigns to improve awareness of cardiovascular disease (CVD) risk in many regions, CVD remains a global concern. The prevalence of cases of CVD increased from 271 million in 1990 to 523 million in 2019, while the years lived with disability due to CVD doubled (from 17.7 million to 34.4 million) [1]. The regional incidence of CVD varies. For instance, in 2020, the CVD prevalence among USA adults aged 20 years and above was 48.6%, accounting for 127.9 million people [2]. While the global prevalence of CVD ranged between 5,881.0 and 11,342.6 per 100,000 in 2022, it was almost double (9,943.9 per 100,000) in western Sub-Saharan Africa (SSA) [3]. In the same year, the overall mortality burden of CVD in SSA ranged between 187.8 and 464.6 per 100,000 people [3]. This calls for urgent action to look for ways of reducing CVD risk in SSA, including Nigeria.

The current prevalence of CVD in Nigeria was as high as 76.11% and a mortality rate of 10%, with an age-standardized incidence rate of 100 to 149.99 per 100,000 persons among adolescents and young adults [4, 5]. In 2019, the age-standardized disability-adjusted life years (DALYs) due to CVD in Nigeria ranged between 4320 and 5790 per 100,000 persons [1] and increased to 5250 per 100,000 persons in 2022 [3]. Many risk factors have been implicated to be responsible for this increased incidence of CVD in Nigeria. These factors include hypertension (30.6%), overweight/obesity (25.5%/14.4%), diabetes mellitus (3.6%), high cholesterol (3.2%), cigarette/tobacco use (male = 5.6%, female = 0.3%, age = 15–49 years), physical inactivity (62.2%), indoor smoke pollution (7.4% rural, 6.8% urban), and unhealthy diet (74.8%) [4]. Measures have been put in place to combat this rising prevalence and burden. These measures include treatment of common non-communicable diseases (NCDs) such as hypertension and diabetes at the primary health care level, availability of national surveillance and monitoring for NCDs, and involvement of NGOs and civil society in policy formulation and implementation, among others [4]. These measures have not yielded the expected results, probably because of the weak and inefficient health system, the lack of national guidelines for the prevention, treatment, and control of many CVDs, and the unavailability of national data on many NCDs [4]. These might have impacted CVD risk perception negatively among community-dwelling adults in Nigeria.

This study was anchored on the health belief model (HBM) [6], which postulated that a low health risk perception (such as low perceived susceptibility, low perceived severity, low perceived benefits, inadequate cues to action, low self-efficacy, and high perceived barriers) might lead to inadequate behavioural changes, such as lifestyle modification for CVD risk reduction. We conceptualised the risk perception on six constructs of HBM: Perceived susceptibility (when community dwellers consider their inappropriate lifestyle might lead to undesirable

outcomes like CVD), perceived *severity* (based on the known fact of the relationship between inappropriate lifestyle and CVD and death), perceived benefits (the expected advantages following appropriate lifestyle modification, such as cardiovascular event risk reduction), perceived barriers (the reluctance to undergo the screening because of difficulty scheduling, lack of funds, and anxiety related to possible abnormal findings), *self*-efficacy (the ability to adhere to the lifestyle modification practices, such as eating a balanced diet, exercising regularly, and adopting healthcare workers' behavioral advice), and cues to action (a close friend who recently suffered an acute CVD after a lifelong inappropriate lifestyle). Studies have shown that perceptions of CVD risk differ among populations [7–10], with most studies reporting a poor perception of CVD risks [8, 10–13]. The prevalence of poor CVD risk perception ranged from 54.1% to 90%, especially in low- and middle-income countries, including SSA [8, 14, 15]. In Nigeria, the prevalence of poor risk perception ranged from 54.1 to 78.1% [10, 13].

In line with the HBM [6], it has been recommended that the CVD risk perception in a population should first be explored before developing and implementing CVD prevention and management strategies [15, 16]. Studies have reported that CVD risk perception significantly predicts health behaviour modification as people who underestimate their risk of developing CVD may not take adequate preventive measures to protect themselves and others [17–20]. Furthermore, a poor CVD risk perception can delay seeking early intervention even in the presence of symptoms [21] and reduce treatment adherence [22]. Therefore, exploring the CVD risk perception of community-dwelling adults will help in targeted campaigns to change poor perception.

There is a dearth of representative population-based studies on the CVD perception of community-dwelling adults in Nigeria. To our knowledge, two studies have reported the perception of CVD among undergraduate students and community dwellers in Ibadan [10, 13]. However, the study samples may not accurately represent the entire southwest population as the studies were conducted in one city. Therefore, this study aimed to assess the CVD risk perception and sociodemographic correlates among rural and urban community dwellers in southwest Nigeria. Specifically, the following research questions were answered: (1) What is the level of CVD risk perception among the participants? (2) Are there any significant differences in CVD risk perception across sex, area (urban and rural), and state of residence? (3) Are there any significant associations between CVD risk perception and participants' age, sex, level of education, marital status, area, and state of residence? (4) What are the sociodemographic adjusted predictors of a good CVD risk perception?

## Methods

### Study design

The study employed a convergent parallel mixed-methods design involving concurrent quantitative and qualitative data collection and analysis. The quantitative survey was a cross-sectional survey using a standardised questionnaire, while the qualitative was a focus group discussion (FGD). The approach provides a means to triangulate and cross-verify the findings from the quantitative and qualitative methods [23].

### Setting

The study was conducted in three of Nigeria's six southwest states: Oyo, Lagos, and Ogun. The two selected communities in each state were (1) Oluyole and Egbeda in Oyo State, (2) Lagos Island and Mojoda in Lagos State, and (3) Sagamu and Iperu in Ogun State. The urban areas are Oluyole, Lagos Island, and Sagamu, while Egbeda, Mojoda, and Iperu are rural. The Yoruba ethnic group dominates Southwest Nigeria, while English is the official language. The

people's occupations are farming, trade, and industry, and their native diet is rich in carbohydrates and meat protein.

## Study participants selection

The participants were community-dwelling adults who were included if they were 18 years and above, could communicate in English or Yoruba, and had no history of CVD diagnoses. Participants were asked if they were told to have or had received treatment for CVD to screen them for the presence of CVD. We employed a multi-stage sampling technique by selecting three states in southwest Nigeria and two communities (one urban and one rural) in each state. The first stage was the selection of three states out of six in southwest Nigeria through balloting without replacement. In each state, the local governments were stratified into rural and urban areas. One community was purposefully selected from each stratum given three urban and three rural communities, ensuring sociocultural representativeness of the overall sample, such as dietary patterns, availability of amenities, religion, social class, and education. Individual participants were randomly selected from public places in each community, such as worship centres (Churches and Mosques). Churches and mosques with a higher population and easy accessibility were chosen in each selected community. The religious leaders in each community helped in facilitating contact with prospective participants, and that enabled easy access to capture their data on the day of worship. Consenting adults who met the eligibility criteria were recruited for the quantitative part. For the qualitative aspect, individuals were purposively recruited using a maximum variation sampling approach. Those who participated in the quantitative part did not participate in the qualitative study. The participants were recruited between January and December 2023.

## Sample size estimation

A population-based sample size formula $Z^2_{crit} P (1-P)/e^2$, was used to estimate the required sample size, given a margin error (e) = 0.05, an estimate of the proportion of the outcome in the previous study (P) = 54.1%, and Z = 1.96 [13, 24]. The formula yielded a minimum of 382 participants in each state. Therefore, a minimum of 1,146 participants from the three states was adequate for 95% statistical power. Thus, we set out to recruit at least 402 participants in each state (201 each from rural and urban communities). An average of eight to ten participants for each of the six FGDs was ideal [25].

## Instruments and procedure

The survey and FGD guide included a biodata form that obtained participants' sociodemographic variables: age, sex, marital status, level of education, area, and state of residence.

**Quantitative study.** Participants' CVD risk perception was obtained using the Perception of Risk of Heart Disease Scale (PRHDS). The PRHDS is a 20-item instrument scored on a 4-point Likert scale (1 to 4) with twelve reversed-scored items. The total score ranges from 20 to 80, with a higher score signifying good perception. The PRHDS has an excellent internal consistency (Cronbach alpha = 0.8) [26].

The PRHDS was cross-culturally adapted into the Yoruba language following the procedure described by Beaton et al. [27]. Trained research assistants distributed the English and Yoruba versions of the questionnaires to participants according to their language fluency. The questionnaire was self-administered; however, the research assistants provided further explanation or assistance to those who needed it. Data collected from all the centres were coded and merged into a single password-encrypted spreadsheet.

**Qualitative study.** The instruments for the FGD were a content-validated focus guide, digital audio recorders, stationery, and researchers' field notes [28]. The FGD was conducted in each community simultaneously with the quantitative data collection. Participants were informed about the date, time, and venue in advance, allowing them to prepare. The FGD was conducted in a circular sitting arrangement, enabling participants to see, listen to, and engage each other [28]. The FGDs were moderated in English and/or Yoruba by each state's lead investigators, who are experts in qualitative interviewing. The moderators asked questions using the focus guide and elicited further responses through probing, prompting, and redirecting. The participants were allowed to discuss freely and spontaneously until each question reached saturation. The audio records were transcribed, anonymised, and merged into a single file for qualitative analyses.

## Data analysis

Quantitative data were analysed using the SPSS version 29. The PRHDS with complete item scores was summed in a separate column. The PRHDS score had 14.1% missing values; hence, the missing values were replaced using a 20-iteration multiple imputation method. For bivariate analysis, the PRHDS score was dichotomised into poor perception ($\leq$40) and good perception ($\geq$41) (the cut-off was theoretically determined). For multivariate analyses, the PRHDS scores were tested for normality, homogeneity of variance, linearity, and univariate outliers using skewness analysis, Levene's test, scatter plot, and standardised Z scores, respectively [29, 30]. The multivariate outliers and multicollinearity were determined using Mahalanobis and the tolerance factor approaches [29, 30].

The participants' characteristics and prevalence of CVD risk perception levels were summarised using the frequency, percentage, mean, and standard deviation. Independent samples t-test and one-way analysis of variance (ANOVA) were used to examine differences in CVD risk perception across sex, area (urban and rural), and state of residence. Following a significant one-way ANOVA result, the Tukey pair-wise post hoc results were reported. The Cramer's V and Phi ($\Phi$) were reported for bivariate association between sociodemographic factors and CVD risk perception. Finally, sociodemographic adjusted predictors of CVD risk perception were determined using simultaneous entry multiple linear regression. The alpha level was set at 0.05.

Qualitative data analysis was completed using ATLAS.ti (version 23) software. The transcripts (transcripts in Yoruba were translated to English) were checked for accuracy against audio records, imported into the software, and analysed using a thematic analysis approach. The data were inductively organised into codes by identifying important quotations and data extracts. Two qualitative data analysts coded the interview transcripts; afterward, codes were assigned to quotations based on the objectives guiding the study. Themes were generated from the codebook. The code books were compared to establish the agreement. There was high intercoder agreement between the two coders, which implies that both data analysts were able to identify similar codes and themes consistently. The result strengthened the trustworthiness of the research findings. The generated codes were systematically grouped into categories, subthemes, and themes and reviewed for proper fit. Code trees and direct participant quotations authenticated the study findings.

## Ethical approval

Ethical approval was obtained from the University of Ibadan/ University College Hospital Health Research Ethics Committee (Approval Nos: UI/EC/22/0200), Health Research Ethics Committee of College of Medicine, University of Lagos (Approval Nos: CMUL/HREC/11/22/1110), Olabisi Onabanjo University Teaching Hospital Ethics Committee (Approval Nos:

OOUTH/HREC/549/2022AP) and Alberta Research Information Services (ARISE–Research Ethics Board Pro 00122480). Participants provided the written informed consent after receiving and understanding the detailed study protocol.

## Results

### Participants' characteristics

Table 1 shows the participants' sociodemographic characteristics. Many survey participants (67.5%) were 40 years or older, with a mean±SD age of 46.90±15.65 years; 66.1% were married, and 62.4% were women. Fifty-three participants (women = 28 {52.8%}) participated in FGD, with 25 (47.2%) from rural communities, seventeen from Lagos State, and 18 each from Oyo and Ogun State.

### Response distributions on PRHDS

The survey result showed a mean CVD risk perception score of 44.40±8.07, with 28% of the participants falling into the poor perception category (Fig 1). The item-response distribution of PRHDS is presented in Table 2. About one in every seven participants believed that they are

**Table 1. Participants' sociodemographic characteristics.**

| Parameter | Quantitative study |
|---|---|
| | *n* (%) |
| **Sex** | |
| Men | 560 (37.5) |
| Women | 931(62.4) |
| Chose not to say | 2 (0.1) |
| **Age group** | |
| Young adults (<40 years) | 484 (32.4) |
| Middle-aged adults (40–59 years) | 670 (44.9) |
| Older adults (≥ 60 years) | 338 (22.6) |
| Chose not to say | 1(0.1) |
| **Marital status** | |
| Single | 293 (19.6) |
| Married | 987 (66.1) |
| Divorced | 31 (2.1) |
| Widowed | 164 (11.0) |
| Chose not to say | 18 (1.2) |
| **Education** | |
| No formal | 97 (6.5) |
| Primary | 243 (16.3) |
| Secondary | 411 (27.5) |
| Tertiary | 725 (48.6) |
| Chose not to say | 17 (1.1) |
| **Area of residence** | |
| Rural | 784 (52.5) |
| Urban | 709 (47.5) |
| **State** | |
| Oyo | 474 (31.7) |
| Lagos | 499 (33.4) |
| Ogun | 520 (34.8) |

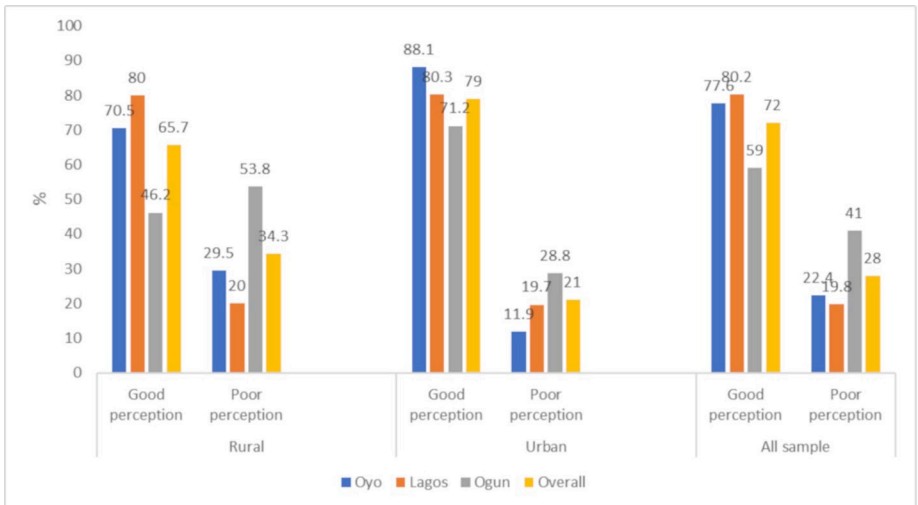

**Fig 1. Proportion of good and poor perception across the state.**

too young to have heart disease (15.5%), people like them do not get heart disease (14.5%), or people of their age are too young to get heart disease (12.7%).

## Comparison of CVD perception score

Table 3 showed that rural residents had a significantly lower CVD risk perception score than their urban counterparts (Mean difference [$MD$] = -3.16, $p < 0.001$). The within-state comparison also showed that rural dwellers had a significantly lower perception score than urban dwellers, except for Lagos state, which is a metropolitan area. There was a significant difference in CVD risk perception scores across the states ($F$ [2, 1490] = 30.523, $p < 0.001$). Post hoc test showed a significantly good perception among Lagos residents compared to Ogun [MD = 3.75, p < 0.001]) and Oyo (MD = 1.07, p = 0.033); and Oyo compared to Ogun ($MD$ = 2.68, $p < 0.001$). There was also a significant difference in perception scores across the states' rural dwellers ($F$ [2, 781] = 41.829, $p < 0.001$), with Lagos having a good perception than Ogun ($MD$ = 6.53, $p < 0.001$) and Oyo ($MD$ = 1.67, $p$ = 0.016), and Oyo higher than Ogun ($MD$ = 4.86, $p < 0.001$).

Men had a significantly good CVD risk perception score than women in Oyo ($MD$ = 1.25, $t$ = 2.009, $p$ = 0.045) and Lagos states ($MD$ = 1.60, $p$ = 0.011). CVD risk perception scores significantly differed across the states' men participants ($F$ [2, 557] = 21.403, $p < 0.001$), such that men in Ogun had significantly poorer perception relative to their counterparts in Oyo ($MD$ = 3.64, $p < 0.001$) and Lagos ($MD$ = 5.00, $p < 0.001$). A similar outcome was observed among women (Table 3).

## Association of sociodemographic characteristics with CVD risk perception

There was a significant bivariate association between sociodemographic factors and CVD risk perception, such that men, compared to women (Phi [$\Phi$] = -0.07, $p$ = 0.012), having tertiary education compared to secondary and less (Cramer's V [V] = 0.18, $p < 0.001$), urban compared to rural dwellers ($\Phi$ = 0.15, $p < 0.001$), and being a Lagos resident compared to other states (V = 0.21, $p < 0.001$) correlate with good CVD risk perception (Table 4).

**Table 2. Response distribution on perception of risk of heart disease scale.**

| S/N | Item | SD (1) f (%) | D (2) f (%) | A (3) f (%) | SA (4) f (%) | Median | Mean |
|---|---|---|---|---|---|---|---|
| 1. | There is a possibility that I have heart disease | 932 (62.4) | 408 (27.3) | 117 (7.8) | 36 (2.4) | 1 | 1.50 |
| 2. | There is a good chance I will get heart disease in the next ten years | 964 (64.6) | 433 (29.0) | 63 (4.2) | 33 (2.2) | 1 | 1.44 |
| 3. | A person who gets heart disease has no chance of being cured | 686 (45.9) | 583 (39.0) | 159 (10.6) | 65 (4.4) | 2 | 1.73 |
| 4. | I have a high chance of getting heart disease because of my past behaviours | 928 (62.2) | 433 (29.0) | 105 (7.0) | 27 (1.8) | 1 | 1.48 |
| 5. | I feel sure that I will get heart disease | 1011 (67.7) | 406 (27.2) | 50 (3.3) | 26 (1.7) | 1 | 1.39 |
| 7. | It is likely that I will get heart diseases | 971 (65.0) | 429 (28.7) | 71 (4.8) | 22 (1.5) | 1 | 1.43 |
| 8. | I am at risk of getting heart disease | 925 (62.0) | 437 (29.3) | 98 (6.6) | 33 (2.2) | 1 | 1.49 |
| 9. | It is possible that I will get heart disease | 938 (62.8) | 438 (29.3) | 89 (6.0) | 28 (1.9) | 1 | 1.47 |
|  |  | **4** | **3** | **2** | **1** |  |  |
| *6. | Healthy lifestyle habits are unattainable | 762 (51.0) | 479 (32.1) | 170 (11.4) | 82 (5.5) | 4 | 3.29 |
| *10. | I am not doing anything now that is unhealthy to my heart | 337 (22.6) | 288 (19.3) | 564 (37.8) | 304 (20.4) | 2 | 2.44 |
| *11. | I am too young to have heart disease | 436 (29.2) | 432 (28.9) | 394 (26.4) | 231 (15.5) | 3 | 2.72 |
| *12. | People like me do not get heart disease | 419 (28.1) | 476 (31.9) | 381 (25.5) | 217 (14.5) | 3 | 2.73 |
| *13. | I am very healthy so my body can fight off heart disease | 274 (18.4) | 333 (22.3) | 585 (39.2) | 301 (20.2) | 2 | 2.39 |
| *14. | I am not worried that I might get heart disease | 328 (22.0) | 341 (22.8) | 529 (35.4) | 295 (19.8) | 2 | 2.47 |
| *15. | People of my age are too young to get heart disease | 494 (33.1) | 491 (32.9) | 318 (21.3) | 190 (12.7) | 3 | 2.86 |
| *16. | People of my age do not get heart disease | 522 (35.0) | 517 (34.6) | 276 (18.5) | 178 (11.9) | 3 | 2.93 |
| *17. | My lifestyle habits do not put me at risk for heart disease | 270 (18.1) | 258 (17.3) | 618 (41.4) | 347 (23.2) | 2 | 2.30 |
| *18. | No matter what I do, if I am going to get heart disease, I will get it | 639 (42.8) | 470 (31.5) | 259 (17.3) | 125 (8.4) | 3 | 3.09 |
| *19. | People who don't get heart disease are just plain lucky | 367 (24.6) | 333 (22.3) | 505 (33.8) | 288 (19.3) | 2 | 2.52 |
| *20. | The causes of heart disease are unknown | 457 (30.6) | 407 (27.3) | 383 (25.7) | 246 (16.5) | 3 | 2.72 |

## Sociodemographic predictors of CVD risk perception score

Similar to the zero-order association, the multivariate regression model showed that having tertiary education (standardised regression coefficient [β] = 0.100, $p < 0.001$), living in urban areas (β = 0.174, $p < 0.001$), and living in Lagos (β = 0.074, $p = 0.013$) and in Oyo other than Ogun state (β = -0.156, $p < 0.001$) significantly predicted having a good perception score

**Table 3. Comparison of CVD risk perception between rural- and urban-dwelling participants and men and women across the states.**

| State | All Mean ± SD | | Rural Mean ± SD | N | Urban Mean ± SD | MD | t-value | p-value |
|---|---|---|---|---|---|---|---|---|
| | | N | | | | | | |
| **Oyo** | 44.98 ± 6.62 | 281 | 43.93 ± 7.15 | 193 | 46.50 ± 5.46 | -2.57 | -4.423 | <0.001* |
| **Lagos** | 46.05 ± 6.67 | 250 | 45.60 ± 6.77 | 249 | 46.49 ± 6.55 | -0.89 | -1.486 | 0.138 |
| **Ogun** | 42.29 ± 9.85 | 253 | 39.07 ± 10.69 | 267 | 45.34 ± 7.86 | -6.27 | -7.592 | <0.001* |
| **All sample** | 44.40 ± 8.07 | 784 | 42.90 ± 8.78 | 709 | 46.06 ± 6.83 | -3.16 | -7.813 | <0.001* |
| **F (df₁, df₂)** | 30.523 (2, 1490) | | 41.829 (2, 781) | | 2.365 (2, 706) | | | |
| **p-value** | <0.001* | | <0.001* | | 0.095 | | | |
| **State** | **All** | | **Men** | | **Women** | **MD** | **t-value** | **p-value** |
| | | N | Mean ± SD | N | Mean ± SD | | | |
| **Oyo** | 44.98 ± 6.62 | 188 | 45.73 ± 6.16 | 286 | 44.48 ± 6.88 | 1.25 | 2.009 | 0.045* |
| **Lagos** | 46.05 ± 6.67 | 169 | 47.09 ± 6.24 | 328 | 45.48 ± 6.83 | 1.60 | 2.554 | 0.011* |
| **Ogun** | 42.29 ± 9.85 | 203 | 42.09 ± 9.80 | 317 | 42.42 ± 9.89 | -0.33 | -0.377 | 0.706 |
| **All sample** | 44.40 ± 8.07 | 560 | 44.82 ± 7.98 | 931 | 44.13 ± 8.11 | 0.69 | 1.590 | 0.112 |
| **F (df₁, df₂)** | 30.523 (2, 1490) | | 21.403 (2, 557) | | 12.147 (2, 928) | | | |
| **p-value** | <0.001* | | <0.001* | | <0.001* | | | |

**Table 4. Association of sociodemographic characteristics with CVD perception.**

| Variable | Perception | | | |
|---|---|---|---|---|
| | **Poor** | **Good** | **Phi** | ***p*-value** |
| | **n (%)** | **n (%)** | | |
| **Sex** | | | -0.07 | 0.012* |
| Men | 136 (32.5) | 424 (39.5) | | |
| Women | 282 (67.5) | 649 (60.5) | | |
| † **Age group** | | | 0.03 | 0.514 |
| Young adults (18–39 years) | 131 (31.3) | 353 (32.9) | | |
| Middle-aged adults (40–59 years) | 184 (44.0) | 486 (45.3) | | |
| Older adults (≥ 60 years) | 103 (24.6) | 235 (21.9) | | |
| **Marital status** | | | 0.02 | 0.433 |
| Do not have a partner | 143 (34.6) | 345 (32.5) | | |
| Have a partner | 270 (65.4) | 717 (67.5) | | |
| † **Education** | | | 0.18 | <0.001* |
| Primary or no formal | 135 (32.4) | 205 (19.4) | | |
| Secondary | 134 (32.1) | 277 (26.2) | | |
| Above secondary | 148 (35.5) | 577 (54.5) | | |
| **Area of residence** | | | 0.15 | <0.001* |
| Rural | 269 (64.4) | 515 (47.9) | | |
| Urban | 149 (35.6) | 560 (52.1) | | |
| † **State** | | | 0.21 | <0.001* |
| Oyo | 106 (25.4) | 368 (34.2) | | |
| Lagos | 99 (23.7) | 400 (37.2) | | |
| Ogun | 213 (51.0) | 307 (28.6) | | |

(Table 5). The model ($F$ [7, 1450] = 22.392, $p < 0.001$) accounted for 9.3% of the variance in perception score (adjusted R square = 0.093).

## Qualitative results

Focus group interviews produced three themes and six sub-themes. The three themes are knowledge about cardiovascular disease, cardiovascular disease risk factors, and prevention of cardiovascular disease.

**Theme 1: Knowledge about cardiovascular disease.** *Subtheme 1.1*: *Knowledge levels*. Participants exhibited a range of awareness regarding CVDs (S1A in S1 File). Basic recognition of CVDs as a serious ailment was noted, and associating it with the heart demonstrated a slightly more nuanced understanding as shown by the following quotes:

> *"What I understand about cardiovascular disease is that disease affecting the heart, basically high blood pressure and what we take in that digests into our system that can cause cholesterol in our body, and it will affect our health, our heart rather."* (OGUN/URBAN)

> *"Cardiovascular disease is a disease of the heart that ehm. . . maybe when somebody has a disease that ehm. . . any disease of the heart ehm. . . it can cause death."* (OYO/URBAN).

One of the participants showcased a detailed comprehension, recognising CVDs as a compound term involving the heart, blood vessels, and the circulatory system:

**Table 5. Sociodemographic predictors of CVD risk perception score.**

| Parameter | Unstandardised Coefficient (B) | Standardised Coefficients (β) | *p*-value | Tolerance |
|---|---|---|---|---|
| Sex (reference = men) | -0.590 | -0.035 | 0.162 | 0.979 |
| Age (unit increase) years | -0.026 | -0.051 | 0.055 | 0.890 |
| Marital status (reference: do not have a partner) | 0.436 | 0.025 | 0.327 | 0.936 |
| Education (reference: secondary and below) | 1.628 | 0.100 | <0.001* | 0.839 |
| Residence (reference: rural) | 2.825 | 0.174 | <0.001* | 0.907 |
| Lagos state (reference: Oyo state) | 1.282 | 0.074 | 0.013* | 0.698 |
| Ogun state (reference: Oyo state) | -2.647 | -0.156 | <0.001* | 0.699 |
| Constant | 44.051 | - | <0.001* | - |

*"It is not only the heart. If you look at it, cardiovascular. It is a medical term that is ah. . . a compound word. We have the heart, cardio, and we have the blood vessels, vascular, and it has to do with the heart per se and all the blood vessels, in company of the blood, the circulatory system, and it enhances the free flow of this functioning and without which a man cannot be said to be in good health"* (OYO/URBAN).

However, some of the participants from rural areas admitted to a lack of knowledge, with responses such as

*"I don't know anything about it. I just used to hear about it, but it has not happened to anyone close to me for me to know how it is." (LAGOS/RURAL)*

*"Ah. . . we don't know it oh. We don't know it oh. You are the one to tell us. God will not allow us to know it oh"* (OGUN/RURAL).

*Subtheme 1.2*: *Information sources*. S1A in S1 File shows that participants gained their knowledge of CVDs through media, personal, or other patients' experiences. For instance, some participants said:

*"I have heard about it very well. I heard about it on the radio."* (LAGOS/RURAL)

*"I think one child that I know that had that kind of experience some years ago, they said he had a hole in the heart. They said it had a hole, and they travelled to go and get it treated, and God helped them to succeed"* (LAGOS/RURAL).

**Theme 2: Cardiovascular disease risk factors.** *Subthemes 2.1*: *Identified risk factors*. The participants also shared their knowledge of CVD risk factors, such as diet-related factors, emotion-related factors, smoking, and obesity (S1B in S1 File).

**Diet-related factors.** *Excessive alcohol use and smoking habits*: Participants perceived excessive alcohol intake and smoking or being exposed to smoke as CVD risk factors.

*"Taking too much alcohol, or taking alcohol also causes heart disease."* (OGUN/URBAN)

*"I want to talk about smoking also; it is also a bigger problem in this society . . .Smoke can cause it"* (OGUN/URBAN)

*Excessive food seasoning and salt intake*: Participants linked too much consumption of cooking salt and artificial seasoning to CVD risk.

*Unhealthy eating habits and poor nutrition choices*: The participants associated CVD risk factors with eating habits and dietary choices. The mention of eating junk food and consuming food without knowledge of its preparation further emphasises the role of dietary practices in their understanding of cardiovascular risk.

*"The kind of eating habit like if you don't eat balanced eh. . . diet food or if you don't eat balanced diet food, it can cause it"* (LAGOS/URBAN)

*"Like not taking a balanced diet, too much of ehm. . . carbohydrate, too much of ehm. . . soft drink or . . . That is what I mean by poor diet"* (OYO/URBAN)

**Emotion-related factors.** Participants perceived that psychoemotional factors such as malice, stress, anger, and anxiety were associated with the risk of developing CVD.

*Malice*: Participants revealed their awareness of risk factors for cardiovascular diseases associated with malice.

*"Someone that does not express his mind and just stores everything he experiences in the memory card of his heart. Your husband committed an offence; you did not ask properly and kept it inside. This one offended you, landlord too offended you, you store it"* (LAGOS/RURAL)

*Stress*: Participants shared their understanding of cardiovascular disease risk factors associated with stress.

*"When I got to the doctor, we were told that what can cause such disease is when one is thinking, when one is thinking about this, or overworking, overlabouring"* (LAGOS/RURAL)

*"What I can add to it is lack of rest as well. When one is passing through a lot of stress, it can trigger cardiovascular disease,"* (OGUN/URBAN)

*Anger*: Participants provided insights into their awareness of cardiovascular disease risk factors associated with anger.

*"Someone that gets angry easily, it can cause it"* (LAGOS/RURAL)

*"Small thing, little thing pissed you up, you get angry, it can also affect the heart and also cause that disease"* (OYO/URBAN)

*Anxiety*: Participants equally provided insights into their awareness of cardiovascular disease risk factors associated with anxiety.

*"Thinking too much can cause cardiovascular disease and things like that"* (OGUN/RURAL),

*"Not only that, when you get anxious too easily. it can also affect the heart."* (OYO/URBAN)

**Other factors.** Moreover, some participants opined that other factor (asides the psychoemotional factors) such as overdose resulting from polypharmacy and unhealthy weight are CVD risk factors.

*Overdose*: In addition, participants expressed awareness of cardiovascular disease risk factors associated with overdose.

*"Overdose of drugs also, I think can cause it. Damage the heart."* (LAGOS/RURAL)

*"When someone is sick and has medications prescribed and instead of taking two, he decides to take four, that thing will cause a problem in his body"* (OGUN/URBAN)

*Overweight and Obesity*: Participants highlighted a potential risk factor for cardiovascular diseases by simply stating,

*"Excessive feeding is too much as well. At times, somebody is getting obese."* (OYO/URBAN),

*"Obesity too because the heart will try to pump to circulate. . . the work the body is doing. . . the heart will not be able to pump very well."* (OGUN/URBAN).

*Subtheme 2.2*: *Perceived dangers of having cardiovascular disease.* A concept diagram for the participants' perceived dangers of CVD was presented in S1C in S1 File.

**Heart failure.**   Participants underscored the potential peril associated with CVD, succinctly expressing,

*"It leads to heart failure"* (/LAGOS/URBAN) and

*"It can lead to failure of the heart."* (LAGOS/URBAN)

These responses emphasise the community's awareness of the severe consequences linked to cardiovascular health issues, particularly the risk of heart failure.

**Sudden death.**   Participants underscored the potential hazards associated with CVD, expressing that:

*"A person that has cardiovascular disease is not far from death if he does not have money to get treated or he has no supporter or anyone to rely on. He is not far from death."* (LAGOS/RURAL),

*". . .a pastor that just died suddenly in the middle of the night because he was not breathing well, and this is due to this cardiovascular disease"* (OGUN/URBAN)

**Stroke.**   Participants identified stroke as a potential danger associated with CVD. The quotes given below reflect an awareness of the connection between cardiovascular issues and the increased risk of stroke.

*"It could lead to stroke as we have said. It could lead to heart attack. It could also lead to diabetes. It could lead to CKD."* (OGUN/URBAN)

*"it always involves hypertension, and hypertension is one of the risk factors for stroke. So, someone. . . if hypertension is not taken care of early, it could lead to stroke."* (OGUN/URBAN)

**Dyspnea.**   Participants equally identified dyspnea (difficulty in breathing) as a danger associated with cardiovascular disease.

*"The person will not be able to breathe well"* (OGUN/RURAL), and

*"Maybe ehn. . . lack of, lack of breathing, the person cannot breathe properly,"* (OYO/URBAN)

**Theme 3: Prevention of cardiovascular disease.** *Sub-themes 3.1*: *Perception of vulnerability to cardiovascular disease*. Participants expressed a particular perception of vulnerability to CVD (S1D in S1 File), stating that:

*"There is no one that cannot have it. There is nothing like a person cannot have it."* (LAGOS/RURAL)

*"The little I can say is that it is a silent killer. Many people are having it, and they don't know that they are victims of cardiovascular disease."* (OYO/URBAN)

These responses reflect an understanding that CVD is not exclusive to certain individuals; rather, everyone is perceived to have the potential to develop it.

*Subthemes 3.2*: *Identified steps to prevent cardiovascular disease*. In light of the foregoing, the participants identified CVD preventive measures (S1E in S1 File).

**Abstinence from identified causes of CVD.** The participants' responses indicate a community awareness of specific actions to prevent cardiovascular diseases through abstinence. One respondent emphasised a general call to abstain from all known causes.

*"Ah. . . Since we know the causes of what happened, we have to abstain from all."* (LAGOS/RURAL)

Another respondent provided information on reducing salt intake and managing emotional stress.

*"Daddy has said, he said when someone is having this high eh. . . annoyance. . .hen en, abstain from it, intake of salt, reduce it and the likes. So. . .abstain from all such."* (LAGOS/RURAL)

**Abstinence from smoking and excessive alcohol intake.** Participants identified abstinence from smoking as a step to prevent cardiovascular disease. They reflected on the deleterious effects of tobacco and excessive alcohol on one's cardiovascular health. It equally shows community awareness of the harmful effects of smoking on both individual and population health.

*"Then smoking too, for those of us that are still smoking, it is dangerous to our health, and we are endangering other persons too"* (OGUN/URBAN)

*"Alcohol eh. . . if we can stop it, abstinence from alcohol."* (OYO/URBAN).

**Adherence to medical prescription.** Further responses from the participants highlighted the significance of adherence to medical prescriptions, including dietary recommendations and prescribed medications, as a proactive measure in preventing cardiovascular diseases.

*"When I went to the hospital, all the things I was told not to eat, I adhered to everything. . ."* (LAGOS/RURAL)

*"Avoid self-medication"* (OYO/URBAN) and

*"If he is able to get medications, and the things he has been told not to do, like if he is told to stop taking alcoholic drinks and he stops taking them, and he keeps taking his medications"* (OYO/RURAL)

**Adequate rest.**   Participants expressed that adequate rest and sleep are essential to preventing cardiovascular disease. Reflect an understanding of the importance of restful sleep and stress reduction for cardiovascular health, particularly for individuals aged 40 and above.

*"Yeah. . . first and foremost, we have to get enough rest and sleep"* (OGUN/URBAN),

*"When you are 40 or above 40, you must not be taking something that is too heavy. You must reduce stress. You rest, you sleep a lot,"* (OGUN/URBAN).

**Seeking medical care.**   Some participants also underscore seeking medical attention as a preventive step. The mention of going to the hospital and consulting with a doctor reflects an understanding of the importance of early detection and professional guidance.

*"When we go to the hospital maybe to see the doctor, and they tell us that a certain thing is wrong with us."* (LAGOS/RURAL)

*"We have to take care of our health by going to the nurses and doctors to examine our BP and the rest."* (OGUN/URBAN).

**Healthy eating habits and balanced diet.**   The participants' responses underscore the community's awareness of the role of diet in preventing CVDs. The emphasis on avoiding foods that can cause heart disease, eating properly cooked meals, avoiding junk, and incorporating fruits and vegetables reflects a proactive approach to dietary choices.

*"It is still about food. The food we eat, we have to change it. We must stop eating the foods that can cause heart disease and start to eat like fruits and vegetables. We will start to eat those ones"* (LAGOS/RURAL)

*"We make sure that ah. . . fruits are not wanting eh. . . in the food we take every day and things that are excessive in oil and fat, we, we, we drop it and we watch what we eat as well."* (OYO/URBAN)

**Regular exercises.**   The participants' responses indicated the community's recognition of the positive impact of regular exercise in preventing cardiovascular diseases. The simplicity of the statements emphasises the direct acknowledgement of the benefits of physical activities.

*"Well . . . exercise is very, very good and helps us a lot to be free from those challenges"* (LAGOS/URBAN)

*"We should also try and exercise our body because exercise reduces the chances of getting ehm . . . CVD, so we should have. . . like there are some things that are trekkable but every time, bike or tricycle, bike or tricycle"* (OGUN/URBAN).

*"You must exercise, which is very key, be it whosoever. No matter your anointing, you must exercise"* (OYO/URBAN)

## Discussion

The study set out to assess the CVD risk perception and sociodemographic correlates among rural and urban community dwellers in southwest Nigeria. In southwest Nigeria, particularly among urban dwellers and men, there is an appreciable level of awareness of the risks associated with cardiovascular disease. Factors such as sex, education level, whether one lives in a rural or urban area, and the specific state of residence significantly influence and predict how people perceive the risk of CVD. This finding is supported by focus group discussions, where participants confirmed their awareness of CVD risks. Their understanding of these risks frequently comes from various sources, including media, interactions with others, and personal experiences. Additionally, many are aware of measures to prevent cardiovascular disease.

The good CVD risk perception exhibited by 72% of our study participants was at variance with many studies that reported low or inadequate risk perception among community dwellers [9, 10, 13–15]. Although these studies estimated risk perception using the same tool as ours, the reason for a good risk perception in the present study may be connected to our participants being apparently healthy community dwellers, unlike some of the previous studies that recruited participants with communicable diseases and CVD. A study has suggested that people with better subjective health status have increased odds of better risk perception [16]. Further reason could be connected to participants' educational level as most of our participants had tertiary education, which could have influenced their knowledge and perception. The participants' level of education reflects the broader population, giving credence to the fact that the adults in southwest Nigeria are educated. The establishment of the first university and the high number of universities in the southwest region may have provided opportunities for people to have tertiary education. Variations in health literacy, cultural differences, or diverse public health messaging across regions could also contribute to the differences. However, our study is similar to 85.8% moderate/high-risk perception among South China adult community dwellers [16]. The medium mean risk perception scores (44.40) in the present study are comparable with scores of college students in Nigeria (44.48) [10] and Turkish adults (48.88) [15]. However, it was lower than that of adults with communicable diseases (53.1) [9] and higher than that of women with CVD (37.15) [14].

It was observed that the area of residence (rural or urban) influenced the risk perception scores. The regression analysis shows that urban community dwellers have increased odds of better risk perception than those in the rural community. The focus group discussion also buttressed this observation as urban focus group participants demonstrated better risk perception. This observation is congruent with a previous study from Nigerian communities, which reported that urban community dwellers have twice the odds of positive risk perception than rural dwellers [13]. Rural college students have also been reported to have lower odds of better risk perception compared with urban colleagues [31]. Urban residents may perceive risk better than rural residents because they have easier access to healthcare, a higher socioeconomic level, and more exposure to health information. The metropolitan nature of urban communities included in the present study may confer this benefit on the participants in urban communities. Better still, their education level is higher, making them better informed. The regression analysis corroborated this assertion, demonstrating that Lagos and Ibadan (Oyo) were strong predictors of risk perception. This finding underscores the urgent need to enhance risk perception, particularly in rural areas, especially among rural dwellers, as better risk perception has been linked with healthy behaviours, lifestyle modification, good health-seeking behaviours, and self-efficacy [17, 19, 20, 32, 33].

The present data shows varied CVD knowledge. The urban community dwellers demonstrate better CVD knowledge, relative to their rural counterparts. A recent study from Nigeria

that used a mixed-method design corroborated this disparity in knowledge about CVD among community dwellers [13]. However, participants in rural and urban communities understand CVD risk factors and its dangers. They identified healthy behaviours and good health-seeking behaviour as means of preventing CVD. This observation is similar to Rwanda's rural and urban community dwellers [34]. Again, there is a need to reinforce this knowledge to promote healthy behaviour in preventing CVD and importantly, among rural community dwellers. Previous studies have suggested that having good knowledge about CVD risk does not necessarily translate to good healthy behaviour or risk perception [33, 35, 36]. This lends credence to the fact that all community dwellers, whether in the rural or urban community, must be targeted for public health educational campaigns to prevent CVD and improve its risk perception [37].

One of the significant findings of this study is the sex association with risk perception in favour of men. Overall, men have a good risk perception score than women. Potential cultural discrimination and psychological and social factors peculiar to women in Nigeria may expose them to unfavourable risk perceptions compared with men. This sex association with risk perception is contrary to the observation of previous studies, which reported no sex association [9, 16, 38, 39]. However, the regression analysis shows that sex does not predict risk perception in agreement with a previous study [40]. Few studies have reported that male sex predicted risk perception [13, 15]. These varied reports about sex-influence on risk perception call for more studies. More specially, as there were large and variable gaps in primary and secondary CVD prevention for women [41]. Women in Nigeria's rural communities likely have lower socioeconomic status, which may influence their risk perception. This was supported by the regression data that showed that a better socioeconomic status predicted better risk perception. Previous studies have confirmed these aforementioned observations. Being an urban dweller, being female, ageing, having higher education, having a higher monthly income, having lower body mass index, having better health status, being non-smoker, and having chronic disease status such having diabetes, higher risk of CVD and family history of CVD were reported to be associated with better CVD risk perception in those studies [11, 15, 16, 31, 40, 42].

This study has some clinical implications. The fact that the prevalence of poor risk perception exceeded a single-digit calls for proactive actions toward improving awareness of CVD risk and its prevention among community dwellers in Southwest Nigeria. We recommend regular public health campaigns and education programs to sensitise people about CVD and healthy behaviours. The public campaign should include how to manage the CVD risk factors, emphasizing regular physical activity, a balanced diet, maintaining proper weight, stopping tobacco use, monitoring blood pressure, glucose, and cholesterol levels, and adequate sleep. Prioritizing educational initiatives targeting adolescents and young adults to emphasize the importance of healthy lifestyle choices and investing in targeted public health campaigns that effectively communicate these long-term health risks could be beneficial. While the measures should be targeted to all community dwellers, special consideration should be given to the socially disadvantaged population. The health care providers can reinforce the awareness campaign during the one-on-one clinical consultation with their clients. These measures may change people's poor perception and ultimately reduce CVD incidence and mortality rates. Equipping healthcare providers to improve CVD risk perception among rural populations will be the right approach. Adequate training of healthcare providers in the evidence-based management of CVD and related risk and appropriate risk stratification will go a long way to encourage behavior change in individual patients. Readiness for task sharing among health workers, community health workers, and treatment supporters will also enhance CVD risk prevention and perception.

## Limitation

This study is not without limitations. Readily available participants were recruited from the churches and mosques using a convenience sampling technique. This might have skewed the results. This technique may have produced samples that do not represent the population. However, the random selection of churches and mosques in each community may have minimised this limitation. Furthermore, we recruited participants for the qualitative study using a maximum variation sampling technique, ensuring the population's representativeness on sociodemographic characteristics. Like other cross-sectional surveys, the participants may have exaggerated or understated their perception. Self-reported tools were employed in the study and may introduce recall bias.

## Conclusion

Few community-dwelling adults had a poor CVD perception, while rural dwellers and Ogun state residents had lower perception scores than their counterparts. Having a secondary education or below and living in rural areas predicted having a low perception score. The results of the focus group discussions showed that participants had varied knowledge and a good perception of cardiovascular diseases as they correctly identified the risk factors and preventive measures. Although some FGD participants admitted limited knowledge of CVD risk, the qualitative results corroborated the quantitative results that showed participants' good CVD perception, indicating the convergence of the two data. The finding of this study suggests strengthening public health in Nigeria, especially in rural areas. It also contributes to emphasizing public campaigns against CVD in low- and middle-income countries.

## Recommendation

We recommend that these gains should be reinforced in public health campaigns and medical outreaches to sensitise people about the disease and promote its early diagnosis. These measures may reduce the CVD prevalence and the mortality rate. Future studies could address how community dwellers use their knowledge of CVD risk and perceptions to prevent CVD.

## Supporting information

**S1 File.** S1a –S1e: Knowledge about cardiovascular disease (S1a), Cardiovascular disease risk factors (S1b), Dangers of having cardiovascular disease (S1c), Perception of vulnerability to cardiovascular (S1d), and Prevention of cardiovascular disease (S1e).
(DOCX)

## Acknowledgments

We thank the leaders of the selected religious centres for granting us access to their members and worship centres. Additionally, we appreciate the contributions of Miss Tolulope Odunaiya, Dr. Iyanuoluwa Odole, and Dr. Moyosooreoluwa Odole from the University of Ibadan Biorepository Laboratory, Dentistry Faculty, and Clinical Science Faculty, respectively.

## Author Contributions

**Conceptualization:** Oluwagbohunmi A. Awosoga, Opeyemi M. Adegoke, Nse A. Odunaiya.

**Data curation:** Oluwagbohunmi A. Awosoga, Olufemi O. Oyewole, Opeyemi M. Adegoke, Adesola C. Odole, Ogochukwu K. Onyeso, Chiedozie J. Alumona, Abiodun M. Adeoye,

Happiness A. Aweto, Blessing S. Ige, Adetola C. Adebayo, Titilope L. Odunaiya, Grace M. Emmanuel, Nurudeen B. Sulaimon, Nse A. Odunaiya.

**Formal analysis:** Oluwagbohunmi A. Awosoga, Olufemi O. Oyewole, Opeyemi M. Adegoke, Adesola C. Odole, Ogochukwu K. Onyeso, Chiedozie J. Alumona, Abiodun M. Adeoye, Blessing S. Ige, Adetola C. Adebayo, Titilope L. Odunaiya, Grace M. Emmanuel, Nurudeen B. Sulaimon, Nse A. Odunaiya.

**Funding acquisition:** Oluwagbohunmi A. Awosoga, Olufemi O. Oyewole, Adesola C. Odole, Happiness A. Aweto, Nse A. Odunaiya.

**Methodology:** Oluwagbohunmi A. Awosoga, Olufemi O. Oyewole, Opeyemi M. Adegoke, Adesola C. Odole, Ogochukwu K. Onyeso, Chiedozie J. Alumona, Abiodun M. Adeoye, Happiness A. Aweto, Blessing S. Ige, Adetola C. Adebayo, Titilope L. Odunaiya, Grace M. Emmanuel, Nurudeen B. Sulaimon, Nse A. Odunaiya.

**Project administration:** Oluwagbohunmi A. Awosoga, Nse A. Odunaiya.

**Supervision:** Oluwagbohunmi A. Awosoga, Olufemi O. Oyewole, Nse A. Odunaiya.

**Writing – original draft:** Oluwagbohunmi A. Awosoga, Olufemi O. Oyewole, Opeyemi M. Adegoke, Adesola C. Odole, Ogochukwu K. Onyeso, Chiedozie J. Alumona, Abiodun M. Adeoye, Happiness A. Aweto, Blessing S. Ige, Adetola C. Adebayo, Titilope L. Odunaiya, Grace M. Emmanuel, Nurudeen B. Sulaimon, Nse A. Odunaiya.

**Writing – review & editing:** Oluwagbohunmi A. Awosoga, Olufemi O. Oyewole, Opeyemi M. Adegoke, Adesola C. Odole, Ogochukwu K. Onyeso, Chiedozie J. Alumona, Abiodun M. Adeoye, Happiness A. Aweto, Blessing S. Ige, Adetola C. Adebayo, Titilope L. Odunaiya, Grace M. Emmanuel, Nurudeen B. Sulaimon, Nse A. Odunaiya.

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
