## [Decision Letter · Decision Letter 0]

17 Sep 2024

PONE-D-24-29870Cardiovascular disease risk perception among community-dwelling adults in southwest Nigeria: a mixed-method studyPLOS ONE

Dear Dr. Oyewole,

Thank you for submitting your manuscript to PLOS ONE. After careful consideration, we feel that it has merit but does not fully meet PLOS ONE’s publication criteria as it currently stands. Therefore, we invite you to submit a revised version of the manuscript that addresses the points raised during the review process.

**ACADEMIC EDITOR: ****Please address all the reviewers' comments. **==============================

We look forward to receiving your revised manuscript.

Kind regards,

Academic Editor

PLOS ONE

Journal Requirements: When submitting your revision, we need you to address these additional requirements. 1. Please ensure that your manuscript meets PLOS ONE's style requirements, including those for file naming. The PLOS ONE style templates can be found at https://journals.plos.org/plosone/s/file?id=wjVg/PLOSOne_formatting_sample_main_body.pdf and https://journals.plos.org/plosone/s/file?id=ba62/PLOSOne_formatting_sample_title_authors_affiliations.pdf 2. Thank you for stating the following financial disclosure: "The study was funded by the Prentice Institute for Global Population and Economy, and the Faculty of Health Sciences, University of Lethbridge, Alberta, Canada." Please state what role the funders took in the study.  If the funders had no role, please state: ""The funders had no role in study design, data collection and analysis, decision to publish, or preparation of the manuscript."" If this statement is not correct you must amend it as needed. Please include this amended Role of Funder statement in your cover letter; we will change the online submission form on your behalf. 3. Please review your reference list to ensure that it is complete and correct. If you have cited papers that have been retracted, please include the rationale for doing so in the manuscript text, or remove these references and replace them with relevant current references. Any changes to the reference list should be mentioned in the rebuttal letter that accompanies your revised manuscript. If you need to cite a retracted article, indicate the article’s retracted status in the References list and also include a citation and full reference for the retraction notice.

Reviewers' comments:

Reviewer's Responses to Questions

**Comments to the Author**

1. Is the manuscript technically sound, and do the data support the conclusions?

Reviewer #1: Partly

Reviewer #2: Yes

2. Has the statistical analysis been performed appropriately and rigorously? 

Reviewer #1: Yes

Reviewer #2: Yes

3. Have the authors made all data underlying the findings in their manuscript fully available?

Reviewer #1: Yes

Reviewer #2: Yes

4. Is the manuscript presented in an intelligible fashion and written in standard English?

Reviewer #1: Yes

Reviewer #2: Yes

5. Review Comments to the Author

Reviewer #1: This study offers valuable insights into the perception of cardiovascular disease risks among community-dwelling adults in Southwest Nigeria. The use of a mixed-methods approach enriches the study, providing both quantitative data and qualitative insights that complement each other. The study is well-organized, with each section contributing to a coherent narrative that underscores the importance of addressing CVD risk perception in public health interventions.

However, there are areas where the study could be strengthened. The introduction could better articulate the research questions and the gaps in the literature that the study aims to fill. The methodology section is detailed but could benefit from more transparency about the sampling process and potential biases. The results are clearly presented, but additional statistical details and qualitative data would enhance the robustness of the findings. The discussion and conclusion are well-argued but could provide deeper analysis and more specific recommendations.

In conclusion, this study makes a significant contribution to the understanding of CVD risk perception in Nigeria, with important implications for public health policy and practice. With some refinements, particularly in the areas of methodology, analysis, and recommendations, the study could offer even greater insights and practical guidance for improving cardiovascular health in the region.

Reviewer #2: Thank you for giving me the opportunity to review this paper. The paper has a nice read, and it was quite enlightening for me in terms of providing knowledge on cardiovascular diseases. it also provides insight into the risk perception of CVD among community dwelling adults. I believe the authors have demonstrated good knowledge of the understand of the study findings. I also appreciate the diagrammatic presentation of the qualitative findings from the thematic breakdown and analysis of the discussion with the participants.

6. PLOS authors have the option to publish the peer review history of their article (what does this mean?). If published, this will include your full peer review and any attached files.

Reviewer #1: No

Reviewer #2: No

---

## [Author Response · Author response to Decision Letter 0]

2 Oct 2024

See the attached response to reviewers' comments.

---

## [Decision Letter · Decision Letter 1]

28 Oct 2024

Cardiovascular disease risk perception among community-dwelling adults in southwest Nigeria: a mixed-method study

PONE-D-24-29870R1

Dear Dr. Oyewole,

We’re pleased to inform you that your manuscript has been judged scientifically suitable for publication and will be formally accepted for publication once it meets all outstanding technical requirements.

Kind regards,

Hilary Izuchukwu Okagbue, Ph.D

Academic Editor

PLOS ONE

Additional Editor Comments (optional):

Reviewers' comments:

Reviewer's Responses to Questions

**Comments to the Author**

1. If the authors have adequately addressed your comments raised in a previous round of review and you feel that this manuscript is now acceptable for publication, you may indicate that here to bypass the “Comments to the Author” section, enter your conflict of interest statement in the “Confidential to Editor” section, and submit your "Accept" recommendation.

Reviewer #1: All comments have been addressed

Reviewer #2: All comments have been addressed

2. Is the manuscript technically sound, and do the data support the conclusions?

Reviewer #1: Yes

Reviewer #2: Yes

3. Has the statistical analysis been performed appropriately and rigorously? 

Reviewer #1: Yes

Reviewer #2: Yes

4. Have the authors made all data underlying the findings in their manuscript fully available?

Reviewer #1: Yes

Reviewer #2: Yes

5. Is the manuscript presented in an intelligible fashion and written in standard English?

Reviewer #1: Yes

Reviewer #2: Yes

6. Review Comments to the Author

Reviewer #1: Initial concerns were adequately addressed, and the authors paid particular attention to queries centering on study integrity (Methodology, results and discussion). Data analysis appeared to be thorough and rigorous. Each section well thought out and written. Overall, study is satisfactory and appropriate corrections have been made to reflect concerns from initial review.

Reviewer #2: All feedback has been incorporated except for the inclusion of Map and the "translation" of transcripts that were transcribed in Yoruba languages (since some of the interviews were conducted in Yoruba language). If a map cannot be found, please include the latter comment and submit again

7. PLOS authors have the option to publish the peer review history of their article (what does this mean?). If published, this will include your full peer review and any attached files.

Reviewer #1: **Yes: **Samuel Onyekweli

Reviewer #2: No

---

## [Editor Report · Acceptance letter]

1 Nov 2024

PONE-D-24-29870R1 

PLOS ONE

Dear Dr. Oyewole, 

I'm pleased to inform you that your manuscript has been deemed suitable for publication in PLOS ONE. Congratulations! Your manuscript is now being handed over to our production team.

Kind regards, 

on behalf of

Dr Hilary Izuchukwu Okagbue 

Academic Editor

PLOS ONE